# Harnessing heterogeneous nucleation to control tin orientations in electronic interconnections

Z.L. Ma [1], S.A. Belyakov[1], K. Sweatman[2], T. Nishimura[2], T. Nishimura[2] & C.M. Gourlay[1]

While many aspects of electronics manufacturing are controlled with great precision, the nucleation of tin in solder joints is currently left to chance. This leads to a widely varying melt undercooling and different crystal orientations in each joint, which results in a different resistance to electromigration, thermomechanical fatigue, and other failure modes in each joint. Here we identify a family of nucleants for tin, prove their effectiveness using a novel droplet solidification technique, and demonstrate an approach to incorporate the nucleants into solder joints to control the orientation of the tin nucleation event. With this approach, it is possible to change tin nucleation from a stochastic to a deterministic process, and to generate single-crystal joints with their c-axis orientation tailored to best combat a selected failure mode.

[1] Department of Materials, Imperial College London, London SW7 2AZ, UK. [2] Nihon Superior Co., Ltd, NS Building, Suita 564-0063, Japan. Correspondence and requests for materials should be addressed to Z.L.M. (email: z.ma13@imperial.ac.uk) or to C.M.G. (email: c.gourlay@imperial.ac.uk)

The nucleation of crystals is a key step in numerous processes in biology, earth science, materials engineering, and beyond. Substantial research has focused on understanding and controlling crystal nucleation, for example: to trigger the nucleation of ice crystals in clouds and cause rain[1]; to prevent the freezing of water in the cells of plants and animals[2]; to produce high-quality protein crystals and other biomolecules[3]; to prevent crystallization and promote glass formation[4]; and to minimize the grain size in metal castings and improve mechanical properties[5, 6]. Nucleation is also central to the microstructures of electronic solder joints and plays an important role in determining the reliability of electronic systems[7].

When electronics fail, the culprit is often the solder joints that interconnect the components. Most electronic solder joints contain at least 95% βSn phase to enable soldering at a temperature tolerable to the electronic components, but the βSn must then operate at up to 80% of its melting point while enduring a high current density, mechanical loading, and thermal cycling. Thus, solder joints are often the weakest link in an electronic system and, with the continuous miniaturization of electronics, the challenges of electromigration, thermomigration, thermomechanical fatigue, and mechanical fatigue are becoming ever more crucial[8, 9]. Each of these failure modes is a strong function of the crystallographic orientation of βSn in a solder joint because tetragonal βSn has highly anisotropic properties. For example, the diffusivities of common solutes (e.g., Cu[10], Ni[11], Ag[12]) are up to four orders of magnitude faster along the c-axis than along the a-axis; the coefficient of thermal expansion (CTE)[13–16] is approximately two times higher along c than a; βSn has its highest stiffness along its direction of maximum CTE (the c-axis)[13]; and βSn also has strong plastic anisotropy[17]. There is a consistent agreement that solder joints with the βSn c-axis parallel with the

**Fig. 1** Typical βSn microstructures and orientations in Cu/Sn-3Ag-0.5Cu/Cu joints. **a** EBSD inverse pole figure maps (IPF-Z) of 12 Cu/Sn-3Ag-0.5Cu/Cu joints that were cooled from ~240 °C in either a DSC at 0.33 K s⁻¹ or an industrial reflow oven at 1–5 K s⁻¹. Wireframe unit cells are superimposed on each main orientation using the mean Euler angles of that grain. All joints with multiple grains are solidification twins with a common <100> axis and common {100} plane indicated by cross-hatching on the unit cells. **b** Cyclic twinning in one joint shown by translating the unit cells into the {101} and {301} cyclic twin configurations[35] with the parallel {100} planes and {101} and {301} twinning planes shaded. **c** <100> and <001> pole figures showing the highly variable βSn grain orientations in 33 joints. **d** The distribution of angles between [001]Sn of all βSn grains and the plane of the Cu substrate

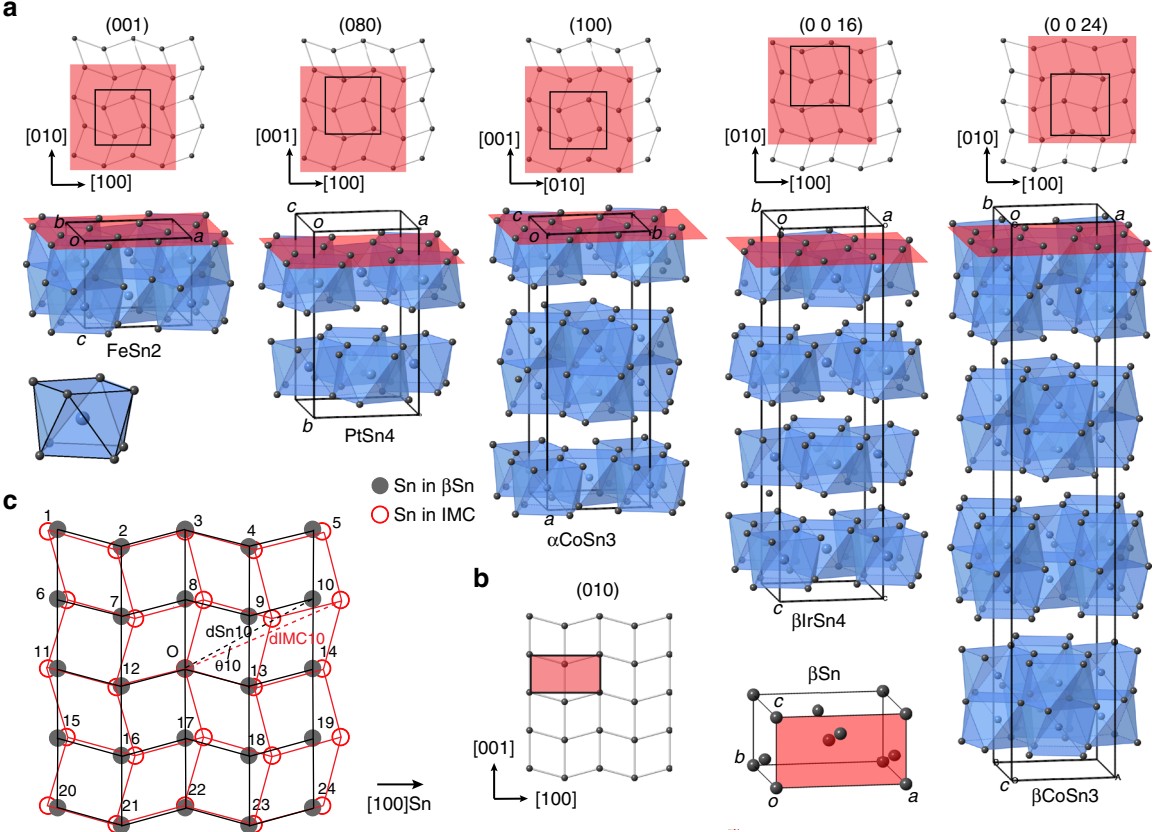

**Fig. 2** Lattice match between βSn and a family of transition metal stannides. **a** Each crystal structure is shown in an equivalent orientation as a stacking of distorted-square antiprisms. A single distorted antiprism with Sn atoms on the vertices and a T atom at the center is shown underneath FeSn₂. Red planes represent the similar net of Sn atoms in each structure. The black square on each net of Sn atoms indicates the projection of the corresponding unit cell, and its position is determined by the origin for the crystallographic settings in Table 1. **b** βSn unit cell and (100) plane. **c** Planar lattice match between the (100) of βSn and the red planes containing Sn nets in the IMCs, showing the atomic misfit of 24 atoms surrounding the origin labeled 'O' (unrelaxed structures). Here αCoSn₃ is used as the example

electron flow direction or temperature gradient suffer the most severe electromigration[18–25] or thermomigration damage[26]. Thermomechanical fatigue is influenced by βSn grain orientations[27] by inducing stresses between adjacent grains of different orientations[27–31], and by inducing stresses between the substrate and solder, which are highest when the βSn c-axis is nearly parallel with the substrate plane as this induces the maximum CTE mismatch[13, 17, 32, 33]. In shear fatigue, joints with the βSn c-axis nearly parallel with the substrate plane and at ~20–60° with the shear direction are more resistant than other βSn orientations[34]. From this, it can be seen that the optimum c-axis orientation is different for different failure modes.

While research continues on the optimum βSn orientations for overall solder joint reliability, it would be beneficial if a method could be developed to reliably control βSn orientations in ball grid array (BGA) and flip chip joints. The challenge is overviewed in Fig. 1, which shows the βSn microstructures and orientations in typical Cu/Sn-3Ag-0.5Cu/Cu BGA joints, which are similar to those widely reported in the literature[13, 18, 35–37, 31, 32]. Figure 1a are electron backscatter diffraction (EBSD) inverse pole figure maps of βSn with respect to the direction of current flow (**Z**). Unit cell orientations are plotted on each grain using the Euler angles of the mean orientation of the grain. It can be seen that some of the joints contain a single βSn grain and some contain two or three βSn grains. The orientations of βSn grains in 33 such joints are plotted relative to the direction of current flow in Fig. 1c. The c-axes, i.e., the <001>, are oriented differently in every joint and are at a wide range of angles from the plane of the substrate

(Fig. 1d). Thus, each joint will be unique in its response to electromigration, mechanical loading, and thermal cycling, and it is likely that an array of joints interconnecting a component will contain at least one joint which is poorly oriented and will act as a weak link.

The joints in Fig. 1 are all single-grain or cyclic-twinned βSn grains, similar to refs [13, 18, 35–37]. The cyclic twinning can be seen from the superimposed unit cells on the two/three grain joints in Fig. 1a that always share a common {100}[35] as indicated by the cross-hatched planes on the unit cells, and from Fig. 1b, which focuses on the cyclic twins in the bottom-right joint from Fig. 1a. Note that the three unit cells have been translated into the {101} cyclic twin and the {301} cyclic twin configurations, which are both consistent with ~60° rotations about a common <100>[35]. Since the joints are all either single grain or twinned, it can be inferred that βSn solidified from a single nucleation event. The location of this nucleation event is one of the Cu₆Sn₅ intermetallic compound (IMC) layers near where the solder ball meets the Cu pad as discussed in ref. [38]. The orientation(s) of βSn in Sn-Ag-Cu solder joints are determined by a stochastic nucleation event and the key to controlling the βSn orientation(s) is to identify methods to tightly control this nucleation event.

Here we develop a novel method to study heterogeneous nucleation mechanisms, where droplets of Sn are solidified on the facets of IMCs and the nucleation undercooling and resulting orientation relationships (ORs) are measured. Combining this method with a lattice matching approach to nucleant design, we identify a family of transition metal stannides that catalyze βSn

**Table 1 Lattice match between Sn net planes of βSn and selected transition metal stannides**

| Phase | Space group | Pearson's symbol | Lattice parameters (Å) | | | Ref. | Sn net plane | $\delta_{<100>Sn}$ (%) | $\delta_P$ (%) |
|---|---|---|---|---|---|---|---|---|---|
| | | | a | b | c | | | | |
| Sn | $I4_1/amd$ | tI4 | 5.831 | 5.831 | 3.182 | 49 | {100} | <100>Sn | – | – |
| AuSn$_4$ | Aba2 | oS20 | 6.512 | 6.516 | 11.707 | 50 | (008) | [100]AuSn$_4$ | 11.7 | 9.4 |
| **PtSn$_4$** | Ccca | oS20 | 6.418 | 11.366 | 6.384 | 51 | (080) | [001]PtSn$_4$ | **9.5** | **8.5** |
| **PdSn$_4$** | Ccca | oS20 | 6.442 | 11.445 | 6.389 | 46 | (080) | [001]PdSn$_4$ | **9.6** | **8.3** |
| **αCoSn$_3$** | Cmca | oS32 | 16.864 | 6.268 | 6.270 | 52 | (600) | [010]αCoSn$_3$ | **7.5** | **6.2** |
| MnSn$_2$ | I4/mcm | tI12 | 6.644 | 6.644 | 5.421 | 53 | (001) | <100>MnSn$_2$ | 13.9 | 11.5 |
| FeSn$_2$ | I4/mcm | tI12 | 6.545 | 6.545 | 5.326 | 54 | (001) | <100>FeSn$_2$ | 12.2 | 10.2 |
| **βIrSn$_4$** | $I4_1/acd$ | tI40 | 6.310 | 6.310 | 22.770 | 47 | (0016) | <100>βIrSn$_4$ | **8.2** | **7.2** |
| **βCoSn$_3$** | $I4_1/acd$ | tI64 | 6.275 | 6.275 | 33.740 | 52 | (0024) | <100>βCoSn$_3$ | **7.6** | **6.1** |

$\delta_{<100>Sn}$ is the linear disregistry between <100>Sn and the parallel direction in each stannide
$\delta_P$ is the planar disregistry defined in Eq. (1) and Fig. 2c. Phases with disregistries < 10% are typeset in bold

nucleation and give useful orientation control. We next demonstrate a simple bonding technique to incorporate the nucleant particles into BGA solder joints to give precisely controlled βSn microstructures and grain orientations tailored to best resist a predefined failure mode.

## Results

**Prediction of nucleant phases.** In order to control the orientation of βSn, we seek seed crystals that can be attached to the pads on the printed circuit board (PCB) side or the component side. The requirements of such a heterogeneous nucleant are: it must be the most potent nucleant in contact with the liquid solder, and generate a useful and reproducible OR with βSn; it must be bondable onto the pad or metallization; it should be a solderable surface when using existing electronic fluxes; and it must be thermodynamically stable in the liquid solder.

Lattice matching has been studied to guide heterogeneous nucleation studies for several decades[39–44], and approaches to identify potent nucleants include the Turnbull–Vonnegut linear disregistry[39], the Bramfitt planar disregistry[40], and the Zhang–Kelly Edge-to-Edge model[41, 45]. The former two approaches are 'plane-on-plane' matching, whereas, in the Edge-to-Edge model, the closest or nearly closest packed planes in the two phases are not necessarily the interfacial plane but their edges meet at the interface and the meeting rows have low linear disregistry[41]. All three approaches indicate that the lower the disregistry, the more potent the nucleation catalyst, and effective nucleation catalysts typically have disregistries of <10%[39–41, 45]. In solders, the tetragonal βSn crystal is more complex than most metals and has a closest packed plane, {010}, that contains zig-zag rows along the closest packed direction, <100>, and straight rows along the next-closest packed direction, <001>, as shown in Fig. 2b. We identify a family of transition metal stannides (IMCs) that are a relatively good lattice match to βSn, both in terms of the closest packed Sn rows and the planar match between the Sn planes in the IMCs to the closest packed {100} in βSn. These compounds are also stable in liquid solder and can be soldered with mild electronic fluxes. Five members of this family of transition metal stannides are shown in Fig. 2a where each has been oriented so that the Sn planes are parallel in each phase. Each of these structure types can be considered as a stacking of TSn$_8$ distorted-square antiprisms (Fig. 2a) with Sn atoms on the vertices and a T atom at the center[46, 47]. Each structure type contains similar planes of Sn atoms (shaded red in Fig. 2a) that are a relatively good planar match to the {010} of βSn. To quantify the lattice match, we take two approaches: first, the disregisty was calculated along the closest packed atomic rows in the interfacial plane, $\delta_{<100>Sn}$, which is the most important factor

emphasized in the Edge-to-Edge model[41, 45] and is also the worst matching closely packed direction in the interfacial plane for all nucleants studied here which makes it a useful single value for the goodness of lattice match in this case.

Second, the planar disregistry was calculated as the average angle-corrected disregistry of 24 atoms surrounding an origin atom as shown in Fig. 2c. This is similar to the Bramfitt approach[40] but involves 25 matching atoms rather than 4 to account for the geometry of this lattice match, and can be calculated as Eq. (1):

$$\delta_{Planar} = \frac{\sum_{i=1}^{24} \frac{d_{IMCi} \times \cos(\theta_i) - d_{Sni}}{d_{Sni}}}{24} \times 100. \quad (1)$$

Disregistry results are given in Table 1 for eight transition metal stannides from the family in Fig. 2. The IMCs which have disregisties <10% calculated by both methods are highlighted in Table 1. These five highlighted phases are potentially good catalysts for βSn nucleation. Among them, PdSn$_4$ has the prototype of PtSn$_4$[46] and similar lattice parameters, and βCoSn$_3$ does not exist in equilibrium at βSn nucleation temperatures[48]. Therefore, PtSn$_4$, αCoSn$_3$, and βIrSn$_4$ were explored in this study.

**Nucleation mechanisms.** As shown in Fig. 3a–c, αCoSn$_3$, PtSn$_4$, and βIrSn$_4$ single crystals all grew with a faceted plate morphology and their largest facets are parallel with the planes of Sn nets in Fig. 2a (i.e., the red planes that are predicted to catalyze βSn nucleation). In Fig. 3a–c, the growth facets and directions have been determined by EBSD as indicated by the unit cells inset in each figure that were plotted from the EBSD-measured Euler angles. To explore βSn nucleation mechanisms on these largest facets of all IMCs, Sn droplets were solidified on each of them, the nucleation undercooling was measured by differential scanning calorimetry (DSC), and the ORs and microstructures were measured by EBSD. Experimental details are as given in the Methods section. Figure 3d shows the typical result of solidifying tin droplets onto the largest IMC facets, using βIrSn$_4$ as an example. The EBSD IPF-Y map and pole figures show that all five tin droplets are single-grain and that the βSn has one of two reproducible orientations with respect to the orientation of the βIrSn$_4$, either [001]Sn‖[100]βIrSn$_4$ (green droplets) or [001]Sn‖ [010]βIrSn$_4$ (yellow droplets) as indicated in the orientation map and pole figures. However, these are just two variants of the same OR since βIrSn$_4$ is tetragonal (i.e., a=b) (Table 1). EBSD maps in all X, Y, and Z directions of these five droplets in Fig. 3d are given in Supplementary Fig. 1. The measured ORs and their frequency of occurrence when tin droplets solidified on the three IMCs are

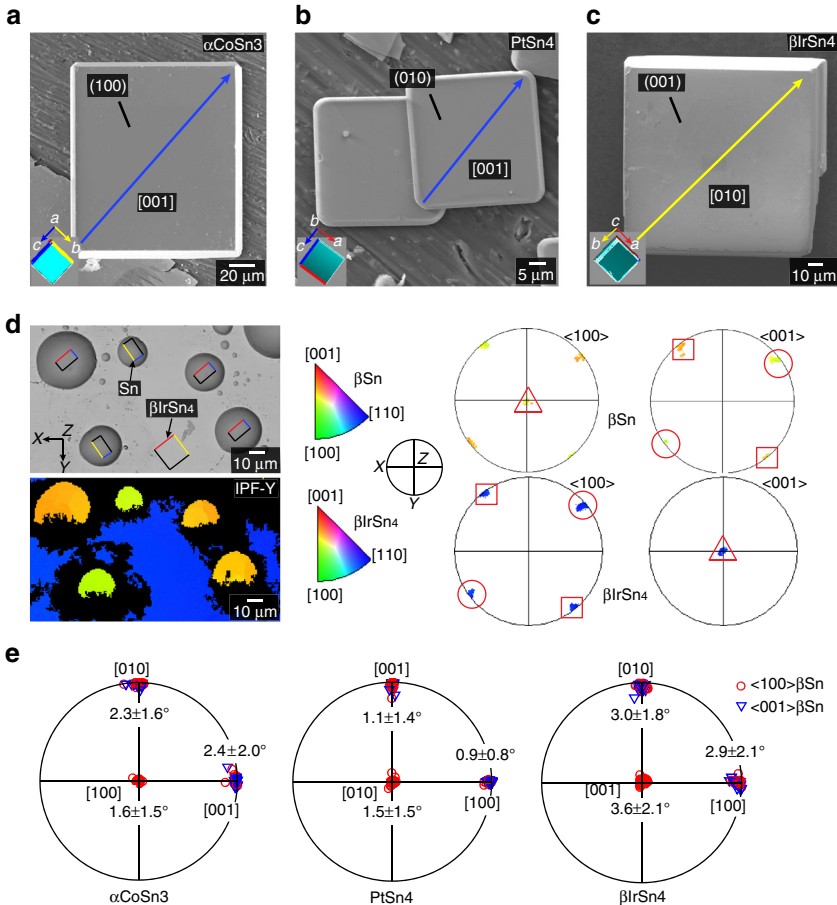

**Fig. 3** Nucleation of Sn droplets on the facets of single-crystal IMCs. **a–c** Typical αCoSn₃, PtSn₄, and βIrSn₄ single crystals. The growth habit is shown in the inserts using unit cell orientations measured by EBSD. The largest facets are parallel to the Sn net plane in Fig. 2a in each phase. **d** A typical example of Sn droplets solidified on the (001) facet of βIrSn₄ and the corresponding EBSD IPF-Y map and pole figures. Each βSn droplet is a single crystal. The superimposed unit cells have a-, b-, and c-axis labeled using the same color as **a–c**. The pairs of triangles, circles, and squares in the pole figures are near-parallel directions in the βSn droplets and the βIrSn₄ crystal. **e** Summarized pole figures of βSn orientations with respect to the largest facets of αCoSn₃, PtSn₄, and βIrSn₄ based on 41, 105, and 36 measurements, respectively. All measured orientations have been rotated towards one equivalent orientation by exploiting symmetry. The angular deviation of each pair of near-parallel planes is given as the mean value and standard deviation

### Table 2 Measured ORs between βSn and IMCs

| Droplets | Measured OR | Frequency of occurrence | $\delta_{<100>Sn}$ (%) | $\delta_P$ (%) |
|---|---|---|---|---|
| αCoSn₃/Sn | {100}$_{Sn}$‖{100}$_{CoSn3}$<001>$_{Sn}$‖<001>$_{CoSn3}$ | 25 out of 41 | 7.49 | 6.16 |
| | {100}Sn‖{100}$_{CoSn3}$<001>Sn‖<010>$_{CoSn3}$ | 16 out of 41 | 7.53 | 6.18 |
| βIrSn₄/Sn | {100}$_{Sn}$‖{001}$_{IrSn4}$<001>$_{Sn}$‖<100>$_{IrSn4}$ | 36 out of 36 | 8.2 | 7.2 |
| PtSn₄/Sn | {100}$_{Sn}$‖{010}$_{PtSn4}$<001>$_{Sn}$‖<001>$_{PtSn4}$ | 12 out of 105 | 10.1 | 8.50 |
| | {100}$_{Sn}$‖{010}$_{PtSn4}$<001>$_{Sn}$‖<100>$_{PtSn4}$ | 93 out of 105 | 9.5 | 8.49 |

The frequencies of occurrence and disregistries of each OR are also given

summarized in Table 2. Noticeably, PtSn₄ and αCoSn₃ each have two reproducible ORs with βSn, but the frequency of occurrence suggests one of these two ORs in each case is more prevalent than the other. This can be understood by noting that PtSn₄ and αCoSn₃ are orthorhombic but they are only slightly distorted from tetragonal (Table 1). Thus, the lattice match is similar in both ORs (see Fig. 2a, b) and, in each case, the more frequently measured OR has a slightly lower (better) disregistry compared with the other OR (Table 2), which indicates the lower disregistry likely results in lower interfacial energy. All measured ORs (Table 2) between each IMC and the βSn are quantified in Fig. 3e as stereographic projections with respect to the largest facet plane

of each IMC. The mean angular difference and the standard deviation of these parallel planes are quantified in Fig. 3e, where it can be seen that all pairs of near-parallel planes had mean angular differences of <4°.

The nucleation potencies of these three IMCs were evaluated from DSC of ~20 μm tin droplets solidifying on the IMC facets. Typical DSC curves and the definition of the nucleation undercooling are given in Fig. 4a, b. Figure 4c shows the nucleation undercooling vs. the lattice match expressed as both the planar disregistry and the linear disregistry along <100>Sn (for nucleants with two ORs, the worse $\delta_{<100>Sn}$ and $\delta_P$ (Table 2) are plotted). The nucleation undercooling is significantly

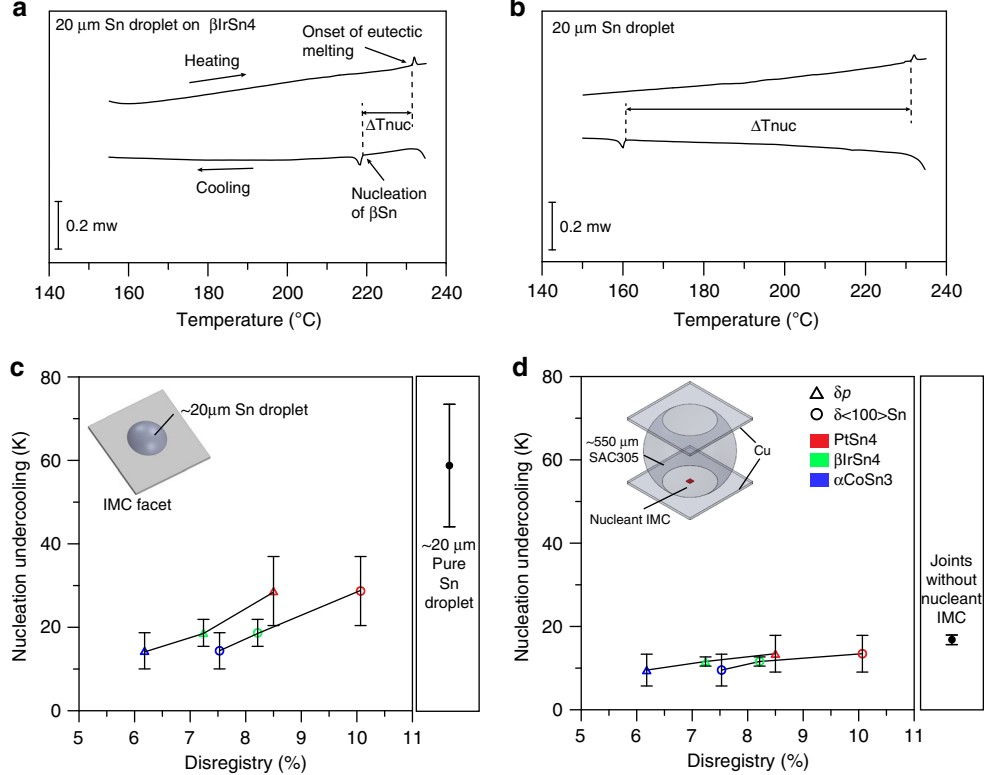

**Fig. 4** Nucleation undercooling vs. lattice mismatch. Typical DSC curves and definition of nucleation undercooling in **a** a single ~20 μm Sn droplet on the (001) facet of a $\beta IrSn_4$ single crystal and **b** a freestanding single ~20 μm Sn droplet. **c** The nucleation undercooling for βSn in ~20 μm Sn droplets on the IMC facets vs. planar disregistry and linear disregistry along <100>Sn. **d** The nucleation undercooling of 550 μm Cu/SAC305 + IMC/Cu solder joints vs. planar disregistry and linear disregistry along <100>Sn. Each datapoint and error bar represent the mean nucleation undercooling and standard deviation from at least 25 measurements. All samples were measured on inert oxidized Al substrates

suppressed compared with the solidification of ~20 μm droplets on inert oxidized Al substrates (58 ± 15 K). It can also be seen in Fig. 4c that the lower the disregisty, the smaller the nucleation undercooling for both methods of quantifying lattice mismatch, consistent with nucleant potency theories[39–41]. At the same time, in Fig. 4c, note that the nucleation undercoolings of ~20 μm droplets on the IMC facets are not very small compared with potent nucleants in other systems (e.g., Al on $Al_3Ti$[44]). However, importantly, the ORs in Table 2 formed in 182 out of 182 droplets and, irrespective of which OR formed, the [001] of tetragonal βSn was always in the plane of the largest facet of the IMC. Thus, these ORs are not only reproducible but also useful since, by controlling the orientation of the largest facet in a joint, these IMCs have the potential to be used as seed crystals to control the orientation of the [001] of βSn in solder joints.

**βSn grain structure and orientation control in solder joints.** Since the natural growth shape of primary $\alpha CoSn_3$, $PtSn_4$, and $\beta IrSn_4$ crystals are thin plates (Fig. 3a–c) whose main facet is the desired nucleation plane, seed crystals were obtained by dissolving the matrix βSn and using the largest natural growth facet as a seed crystal without the need to take slices from a wafer of IMC. To incorporate a $PtSn_4$, $\alpha CoSn_3$, or $\beta IrSn_4$ seed crystal into BGA solder joints, they were first bonded to Cu pads using a form of transient liquid phase bonding (TLPB), where the IMC seed crystal remained solid during the TLPB. An immersion tin coating was applied to Cu pads and the nucleant IMC was laid flat with the main facet in the plane of the pad as shown in Fig. 5a using $\alpha CoSn_3$ as an example. The $\alpha CoSn_3$ was then TLPB to the pad by a reflow of 5 min at 240 °C, which resulted in the Cu/

$Cu_3Sn/Cu_6Sn_5/\alpha CoSn_3$ layers shown in Fig. 5b. Since $PtSn_4$, $\alpha CoSn_3$, and $\beta IrSn_4$ are all solderable surfaces using a standard ROL-1 flux, these nucleant-modified pads were then used in the same manner as a Cu-OSP substrate and solder joints were made following the procedure in the Methods section. Figure 5c shows the cross-section of a typical Cu/Sn-3.5Ag + IMC/Cu solder joint after double reflow where the bottom Cu pad contains a TLPB $\alpha CoSn_3$ seed crystal and a 550 μm Sn-3.5Ag ball was used. The microstructure consists of primary $Cu_6Sn_5$, βSn dendrites with <110> growth directions indicated by the blue arrows, βSn + $Ag_3Sn + Cu_6Sn_5$ interdendritic eutectic, and a bonded $\alpha CoSn_3$ particle on the bottom substrate. Note that a large $\alpha CoSn_3$ seed crystal has been used in this example to make it feasible to polish to a cross-section containing the seed crystal and measure the OR. The EBSD IPF-Z map of the βSn phase is shown in Fig. 5d, which indicates that the joint contains a single crystal of βSn. Figure 5e is the EBSD IPF-Z map of the $\alpha CoSn_3$ phase. In both maps, the unit cell wireframes of these two phases are superimposed to show the OR, which is also indicated by pole figures in Fig. 5f where near-parallel directions are indicated with circles, squares, and triangles. The OR is consistent with that in Table 2 from droplet studies. Due to the special placement of the $\alpha CoSn_3$ particle, the nucleated single βSn grain is oriented with [100] or [010] across the joint and with [001] parallel with the substrate plane, i.e., the c-axis of βSn is in the substrate plane. Note that, even though primary and interfacial $Cu_6Sn_5$ are present prior to βSn nucleation, the nucleation of βSn always occurred on the seed crystal because they ($\alpha CoSn_3$, $PtSn_4$, $\beta IrSn_4$) are more potent nucleants than $Cu_6Sn_5$. Primary $Ag_3Sn$ plates were not observed here because the seed crystals require only a relatively small undercooling for βSn nucleation. Further examples of the phases

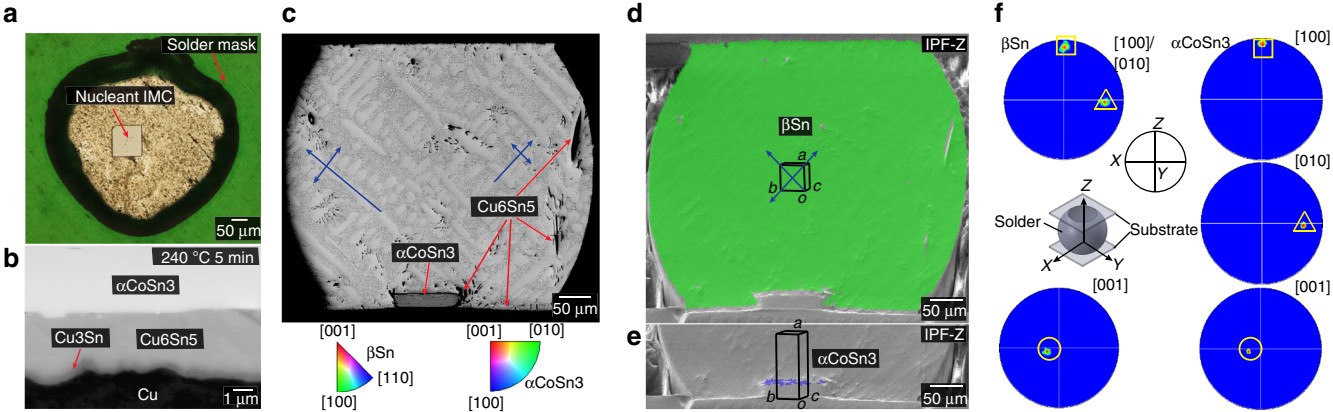

**Fig. 5** Microstructure at different stages of processing in a Cu/Sn-3.5Ag + nucleant/Cu joint. **a** A typical immersion tin-coated Cu substrate with an IMC nucleant bonded on top. **b** Cross-section of a typical transient liquid phase bonded (TLPB) αCoSn$_3$ seed crystal (i.e., a cross-section through **a**). **c** Cross-section of a typical Cu/Sn-3.5Ag + αCoSn$_3$/Cu solder joint after double reflow. Blue arrows show <110> βSn dendrite branching directions. **d** EBSD IPF-Z map of the βSn phase. **e** EBSD IPF-Z map of the αCoSn$_3$ seed crystal. The unit cell orientation of each phase is superimposed on each map. **f** Pole figures of the βSn and αCoSn$_3$ phases. The pairs of triangles, circles, and squares indicate the near-parallel planes that are consistent with the OR in Figs. 2c, 3

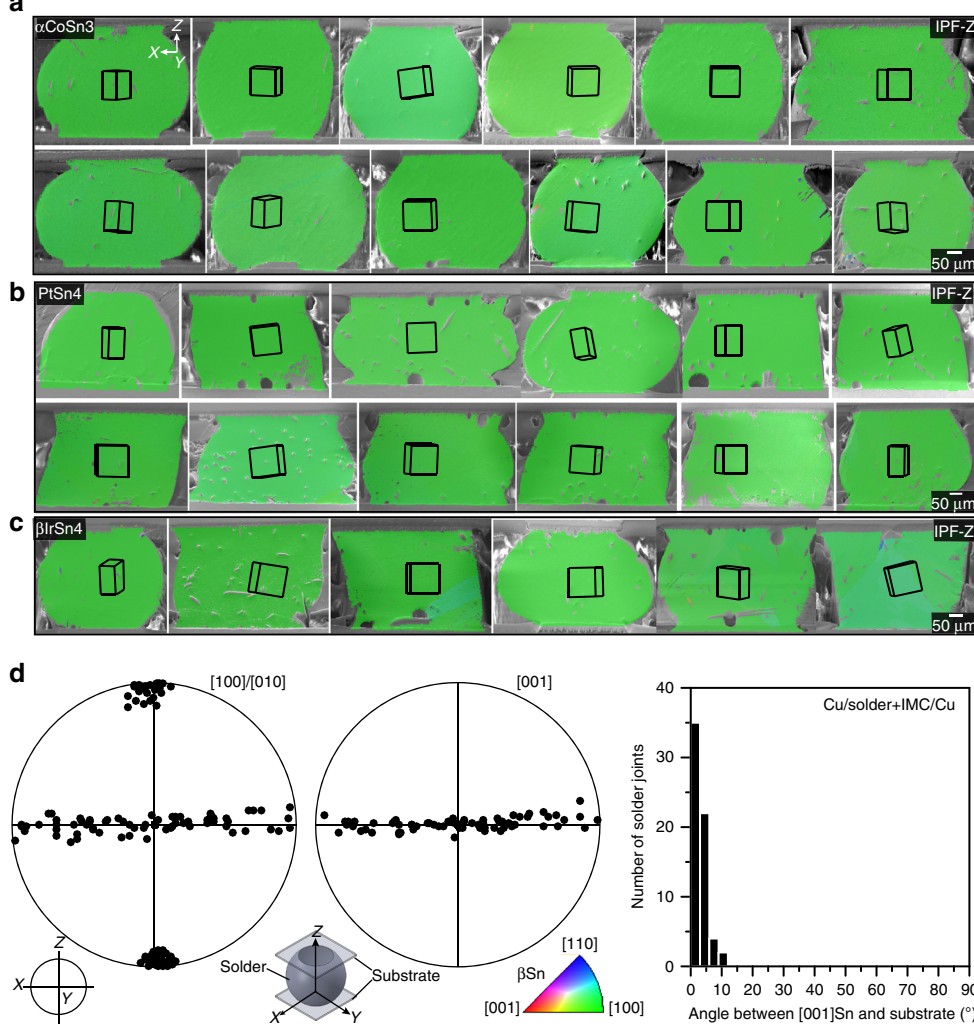

**Fig. 6** c-axis orientation control in Cu/Sn-3Ag-0.5Cu or Sn-3.5Ag + nucleant/Cu joints. **a–c** Typical EBSD IPF-Z maps and unit cell wireframes. **a** αCoSn$_3$, **b** PtSn$_4$, and **c** βIrSn$_4$. **d** Summarized βSn pole figures of 67 solder joints and a histogram of the angles between [001]Sn of these joints and the substrate plane. This range of orientations is optimum for resisting electromigration according to refs[18-25]

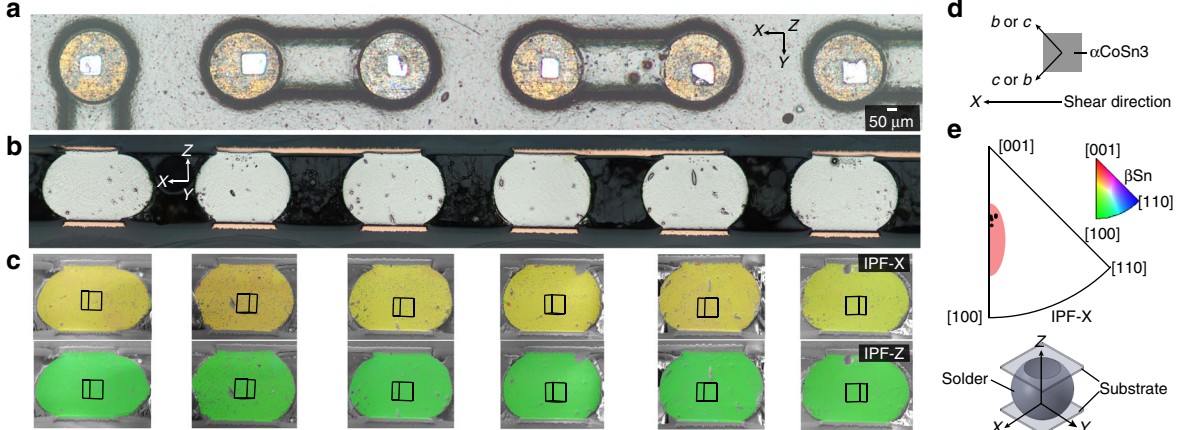

**Fig. 7** Combined *c*-axis and *a*-axis orientation control in Cu/Sn-3.5Ag + αCoSn₃/Cu joints. **a** αCoSn₃ particles transient liquid phase bonded onto an array of immersion tin-coated Cu pads. All seed crystals are aligned to have their *c/b*-axis in the *XY* plane and at ~45° with the *X* direction. **b, c** Cross-sections of Cu/Sn-3.5Ag + αCoSn₃/Cu joints after double reflow on the seeded substrates in **a**. **b** Optical micrographs. **c** EBSD IPF-X and IPF-Y maps. **d** Schematic of the orientation of the αCoSn₃ particle with respect to the coordinate system and the shear direction. **e** IPF-X of all solder joints in **b, c**. The red region is the range of orientations reported to give the best shear fatigue performance in ref. [34]

are shown in Supplementary Fig. 2. EBSD maps in all *X*, *Y*, and *Z* directions of the joint in Fig. 5c are given in Supplementary Fig. 3.

Similar results to Fig. 5 were obtained using PtSn₄ and βIrSn₄ as seed crystals, although a longer time and higher temperature (180 min at 300 °C) were needed to TLPB these IMCs to Cu pads. Figure 6a–c are 30 typical EBSD orientation maps (IPF-Z) of Cu/solder + IMC/Cu joints made with αCoSn₃, PtSn₄, or βIrSn₄ seed crystals and either Sn-3.0Ag-0.5Cu or Sn-3.5Ag solder. The IMC seed crystals have a wide distribution of sizes from ~20–150 μm (length of the longest edge of the plate) and some are TLPB to the upper Cu substrates and others to the lower Cu substrate. In all cases, the IPF-Z maps show a single green *z*-orientation, indicating a single βSn grain with a <100>Sn across the joint and <001>Sn in the plane of the substrate in every joint. For those joints for which the seed crystals are in the sectioning plane, the seed crystal-βSn OR is the same as in Table 2 in every joint. EBSD maps in all *X*, *Y*, and *Z* directions of these joints in Fig. 6a–c are given in Supplementary Figs. 4, 5, and 6.

The βSn orientation in 67 joints are summarized in <100> and <001> pole figures in Fig. 6d as well as the distribution of angles between the *c*-axis of βSn grains and the substrate plane for all 67 joints. All joints show consistent control of the βSn orientation and the *c*-axis of βSn and the substrate plane are parallel to within ~10°. This misorientation between the [001]Sn and the substrate is mostly because the largest facets of the seed IMCs were not perfectly parallel with the substrate after TLPB, especially when the seed crystals are small (e.g., ~20 μm wide). Despite this, Fig. 6 shows that tight control of the orientation of [001]Sn relative to the substrate can be reliably achieved.

The nucleation undercooling of 550 μm Cu/SAC305 + IMC/Cu joints for each type of IMC seed crystal were measured and are plotted vs. the lattice disregistry in Fig. 4b. There is a similar trend with what was measured for tin droplets on IMC facets in Fig. 4a, i.e., the smaller (better) the disregistry the smaller the βSn nucleation undercooling in the joint. The nucleation under-cooling is not significantly suppressed compared with the Cu/SAC305/Cu joint (17 ± 1 K) but nucleation always occurred on the seed crystal giving orientation control. It can also be seen that the nucleation undercooling is smaller for 550 μm BGA joints than for ~20 μm droplets when they solidify on the equivalent facet planes but similar orientation control occurred in both.

## Discussion

The approach in Figs. 5 and 6 gives a *c*-axis orientation perpendicular to the direction of current flow which has been shown to give optimum resistance to electromigration and thermomigration in solder joints in past work[18–25,26]. For some applications, alternative orientations may be preferred and the βSn *c*-axis may need to be tailored to suit the application. For example, it has been shown by Arfaei et al.[34] that shear fatigue performance is best when the βSn *c*-axis is near-parallel with the substrate plane and at ~20–60° angle with the shear direction. This can be achieved with the current approach since the crystallographic orientation of the IMC plates can be readily determined from the macroscopic shape of the plate, and the IMC plates can be pick-and-placed to have a tightly controlled rotation angle. To demonstrate this, in Fig. 7a, αCoSn₃ particles have been bonded along the edge of an array of Cu pads with edges parallel with the *X* direction (the assumed shear direction). This makes the *c/b*-axis of each αCoSn₃ particle at ~45° with the *X* direction, as shown by the configuration in Fig. 7d. Therefore, after double reflow, the final βSn grain orientations in these joints are controlled to have the *c*-axis at ~45° with the *X* direction due to heterogeneous nucleation on the seed crystal. This can be seen in the EBSD IPF-X and IPF-Z maps in Fig. 7c and the summarized IPF-X in Fig. 7e. The red highlighted region in Fig. 7e represents the optimum shear fatigue performance range determined in ref.[34], assuming that *X* is the shear direction.

For applications where thermomechanical fatigue is the dominant issue, the current technique could be altered to fix the nucleant IMC to have the largest facet perpendicular with the substrate plane and one edge parallel with the substrate plane. By doing this, the final joint would have the βSn *c*-axis at ~45° with the substrate plane, which would eliminate the worst situation for the thermomechanical fatigue (i.e., *c*-axis parallel with the substrate[13, 17, 32, 33]). As the understanding of the role of crystal orientation on solder joint performance improves in the future, the approaches overviewed here open the possibility to tailor the *c*-axis orientation to best resist the dominant failure mode.

Here, orientation control has been demonstrated for the first-generation Pb-free solders, Sn-3Ag-0.5Cu and Sn-3.5Ag. The latest generation solders under development commonly additionally contain Ni, Bi, In, etc. It is shown in the Supplementary Fig. 7 that similar orientation control can be achieved in solders with these additions. The seed crystals remain effective with these

additions because they do not react with the seed crystals or introduce a more potent nuclean phase. For example, the Ni addition mostly influences the $Cu_6Sn_5$ phase, which is a less potent nuclean than the seed crystals, and these Bi and In additions introduce extra (Bi) and $\zeta$(Ag,In) phases that form later during solidification and do not strongly affect the nucleation of βSn. The successful orientation control in these Ni-, Bi-, and In-containing solders indicates that this orientation control method is likely to be applicable to solder compositions developed in the future. Thus, it is suggested that the development of solders for optimum reliability could be tackled with a double approach of alloy design and orientation control.

In summary, seed crystals for solder joints create new design opportunities for electronics manufacturing, opening the possibility of designing joints to combat different failure mechanisms and increasing the reproducibility of microstructures across solder arrays. The simple bonding step that incorporates the nucleants onto Cu pads is well suited to pick-and-place technology and the seed crystals are expected to be compatible with future improvements in solder alloy design. More broadly, the droplet solidification technique developed here has the potential to accelerate nuclean discovery in other systems by generating statistically significant datasets on nucleation ORs in a relatively short time.

## Methods

**Synthesis of intermetallic single crystals.** $PtSn_4$, $\alpha CoSn_3$, and $\beta IrSn_4$ single crystals were grown in Sn-rich liquid by cooling hypereutectic Sn-0.2Pt, Sn-0.1Co, and Sn-1Ir (mass%) alloys at $0.33\,K\,s^{-1}$ from 400 °C. The IMC crystals were extracted by selective dissolution of the βSn phase in a solution of either 5% NaOH and 3.5% *ortho*-nitrophenol in distilled water or 50% HCl solution, and were then ultrasonically bathed in ethanol.

**Droplet solidification experiments.** A purity of 99.9% of Sn particles (balls) between 1 and 50 μm were placed on the largest facets of the as-extracted (without further processing) $PtSn_4$, $\alpha CoSn_3$, and $\beta IrSn_4$ single crystals with a $NH_4Cl$-$ZnCl_2$ based flux (Stay-Clean liquid flux, Harris). The Sn was melted on a hot plate with a peak temperature of 240 °C to ensure wetting and spreading on the facet, and then IMC crystals with Sn droplets spread on were ultrasonically bathed in ethanol to remove flux residues. Thermal cycles were then performed in a Mettler Toledo DSC1 under a nitrogen atmosphere with a heating rate of $0.17\,K\,s^{-1}$, peak temperature 240 °C, time above the eutectic temperature of 180 s, and a cooling rate $0.33\,K\,s^{-1}$. The nucleation undercooling was determined from the DSC data and the OR between the βSn droplet and IMC was measured by EBSD.

**BGA solder joints.** Sn-3.0Ag-0.5Cu and Sn-3.5Ag (mass%) alloys were prepared from commercial purity Sn (99.9%), Ag(99.9%), and Cu (99.9%) in a graphite crucible at 500 °C. After 1 h holding, the melts were stirred with a preheated graphite rod and 40 g samples were cast into a chemical analysis mold for X-ray fluorescence spectroscopy analysis. The measured compositions of these alloys are given in Supplementary Table 1. To prepare solder balls with a diameter of $550 \pm 25$ μm, Sn-3.0Ag-0.5Cu and Sn-3.5Ag were rolled to 0.05 mm foils, punched to ∅1.6 mm discs, and reflowed on a non-wetting highly oxidized Ni sheet with an ROL-1 tacky flux (Nihon Superior Co., Ltd) and peak temperature of 280 °C. BGA joints were made using either 99.9% Cu coupons or Cu-OSP (organic solderability preservative) FR4 PCBs as substrates, where both were masked with 500 μm Cu pads. For reflow in a Mettler Toledo DSC1, the following conditions were used: heating rate $0.17\,K\,s^{-1}$, peak temperature 240 °C, time above the eutectic temperature ~180 s, and cooling rate $0.33\,K\,s^{-1}$. For reflow under near-industrial conditions, a forced air convection Tornado LFR400 reflow oven (Surface Mount Technology, Isle of Wight, UK) was used with the following thermal profile: heating rate $1\,K\,s^{-1}$, peak temperature 250 °C, time above the eutectic temperature 80 s, and cooling rate $\sim 3\,K\,s^{-1}$.

**Incorporating nucleants into BGA joints.** $PtSn_4$, $\alpha CoSn_3$, and $\beta IrSn_4$ single crystals were incorporated into BGA joints by two additional steps: (1) Cu pads were coated with a ~1 μm tin layer using a commercial immersion tin solution (Mega Electronics Co., Ltd, UK); (2) $PtSn_4$, $\alpha CoSn_3$, or $\beta IrSn_4$ crystals were placed on the tin layer with $NH_4Cl$-$ZnCl_2$-based flux (Stay-Clean liquid flux, Harris) coated on and their main facet near-parallel with the substrate at a desired rotation angle, and the nucleants were transient liquid phase bonded to the substrate by holding at 240 °C for 5 min (for the $\alpha CoSn_3$ case) or at 300 °C for 180 min (for the $PtSn_4$ and $IrSn_4$ cases). Since $PtSn_4$, $\alpha CoSn_3$, and $\beta IrSn_4$ are all solderable surfaces

using a standard ROL-1 flux, these nucleant-modified pads were then used in the same manner as a Cu-OSP substrate. All solder joints were firstly made by reflowing in the Tornado LFR400 reflow oven. Some joints were reflowed subsequently in the DSC to measure the nucleation undercooling. Substrates were cut to ~$3 \times 3\,mm^2$ for DSC measurements. $N_2$ atmosphere was used in all reflowing cases.

**Data availability.** The EBSD and DSC datasets generated during the current study are available in the Zenodo repository (doi: 10.5281/zenodo.884113).

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

## Acknowledgments

Z.L.M. gratefully acknowledges the China Scholarship Council (CSC) (201306250005) for financial support through the Imperial-CSC scholarship scheme. C.M.G. and S.A.B. gratefully acknowledge funding from the UK EPSRC (grant numbers EP/M002241/1 and EP/N007638/1 (the EPSRC Future LiME Hub)). All authors are grateful for support from Imperial College London- Nihon Superior research project MMRE_P42808.

## Author contributions

Z.L.M., S.A.B. and C.M.G. developed the nucleants; Ta.N., Te.N., K.S. and Z.L.M. optimized methods for incorporating nucleant particles into joints; Z.L.M. performed the experiments; Z.L.M. and C.M.G. wrote the paper with inputs from the other authors.

## Additional information

**Competing interests:** The authors are co-inventors on a patent entitled "Solder joint and bonding method", Japanese patent number: 2017-133073.

