## [Peer Review File · Nature Communications]

Reviewers' comments:

Reviewer #1 (Remarks to the Author):

This paper presents a major step forward for improving the reliability and design of solder joints. It could well be the start of a revolutionary change in the industry. The authors have recognized the importance of oriented nucleation and found a practical means to get this under control, with impressive results.

There are minor editorial suggestions in the attached document, but the paper is publishable in NCOMMS.

Thomas R. Bieler

Attached document:

This paper presents a major step forward for improving the reliability and design of solder joints. It could well be the start of a revolutionary change in the industry. The authors have recognized the importance of oriented nucleation and found a practical means to get this under control, with impressive results. There are minor editorial suggestions below, but the paper is publishable in NCOMMS.

Harnessing heterogeneous nucleation to control tin orientations in electronic interconnections

Z.L. Ma^{1*}, S.A. Belyakov¹, K. Sweatman², T. Nishimura², T. Nishimura², C.M. Gourlay^{1*}

p. 5 line 21 electronic fluxes, not electronics fluxes

The reason for the positioning of the black square in Figure 2A is not apparent.

p. 6: How were Sn droplets made and deposited on the substrate? This is not described. Were they deposited as balls, and then heated and cooled in a DSC?

Figure 4 caption: Please indicate what the 'inert' substrate was in the text as well as the caption or top of part (B) of the figure (e.g. 'glass?').

Also, did you use oriented single crystals of all three of the Stannide IMC materials? This is not clear. There should be enough information so that others can reproduce your work.

p. 8

Al on Al₃Ti₄₄). However, importantly, the ORs reported in Table 2 for all 182 of the droplets examined, irrespective of which OR formed, the [001] of tetragonal βSn was always in the plane of the largest facet of the IMC. Thus, these ORs are not only reproducible but also useful since, by controlling the

You describe facets of the IMC, but the reader has no way of know what the facet refers to. I thought you were using a polished single crystal, but reference to a facet makes this seem like there was a polycrystal. Please clarify what the condition of the substrate was?

β Sn grain structure and orientation control in solder joints. To incorporate a PtSn_4 , αCoSn_3 , or βIrSn_4 seed crystal into ball-grid array (BGA) solder joints, they were first bonded to Cu pads using a form of transient liquid phase bonding (TLPB). An immersion tin coating was applied to Cu pads and the nucleant IMC wafer was laid flat with the main facet in the plane of the pad as shown in Figure 5 (A) using αCoSn_3 as an example. The αCoSn_3 was then TLPB to the pad by a reflow of 5 minutes at 240°C,

Please clarify: while you used TLPB, did the little wafer of the IMC stay solid during the bonding process? Was there any flux used?

Regarding the prior comment, it appears that you sliced wafers of a IMC single crystal and that you refer to the flat surface as the facet. Is this correct?

p. 9

The nucleation undercooling of 550 μm Cu/SAC305+IMC/Cu joints for each type of IMC seed crystal were measured and are plotted versus the lattice disregistry in Figure 4 (D?). There is a similar trend

Practical question – did you put a board+chip with a single solder joint in the dsc? It is hard to imagine what you did to get the data in Figure 4D?

The methods section answers above questions adequately, except for the making of a bga joint in a DSC. In the main body of the text, clarifying how you made and placed the nucleant wafers would help the reader not get distracted from your story.

p. 12

The Sn was melted on a hot plate with a peak temperature of 240 °C to ensure wetting and spreading on the facet, and then flux residues were removed. Thermal cycles were then performed 10

How were flux residues removed?

chemical analysis mould for X-Ray Fluorescence (XRF) spectroscopy analysis. To prepare solder balls with a diameter of 550 ± 25 μm , Sn-3.0Ag-0.5Cu and Sn-3.5Ag were rolled to 0.05 mm foils, punched to $\varnothing 1.6$ mm discs, and reflowed on an inert hotplate with a ROL-1 tacky flux (Nihon Superior Co., 20

what was the inert surface?

p. 13

crystals were placed on the tin layer with their main facet near-parallel with the substrate at a 7 desired rotation angle, and the nucleants were transient liquid phase bonded to the substrate by 8 holding at 240 °C for 5 min (for the αCoSn_3 case) or at 300 °C for 180 min (for the PtSn_4 and IrSn_4 9 cases). Since PtSn_4 , αCoSn_3 and βIrSn_4 are all solderable surfaces using a standard ROL-1 flux, these 10 nucleant-modified pads were then used in the same manner as a Cu-OSP substrate.

What about environment – in air or nitrogen?

Reviewer #2 (Remarks to the Author):

This is a very good paper, well planned, well implemented and well presented. However., I m not sure that it is suitable for publication in Nature. My understanding is that Nature papers should present significant new science and, while the implementation of the ideas presented here is new and definitely contribute to solder technology, I really don't see any new science. The basic ideas are well known, long discussed, and widely used in one form or another.

The paper is good and interesting, and I will be happy to recommend its publication in Nature Communications if the editors believe its scientific (as opposed to technological) novelty is sufficient to justify it.

Reviewer #3 (Remarks to the Author):

Both academic and industrial people are eager to seek methods to effectively control the grain orientation of solder joints due to the anisotropy of β -Sn in mechanical and physical properties. The authors claim they have generated single crystal joints by using IMC nucleants as seed crystal. The authors are supposed to address the comments:

1. For Fig. 3, could the authors provide the EBSD map in X and Z directions? Similarly, the EBSD maps in X and Y directions for Figs. 5 and 6.
2. Several Sn-based solder alloys have been used in this study. Usually for Sn-Ag and Sn-Ag-Cu solders, Ag_3Sn and Cu_6Sn_5 IMCs are the primary phases rather than β -Sn dendrites during a conventional reflow process. The precipitation of these IMCs also have a great effect on the nucleation of β -Sn. Did the author observe the formation of Ag_3Sn particles or plates in the Ag-bearing solder joints? If not, where is Ag? If yes, did the IMC phases still precipitate prior to β -Sn? Also, since the solubilities of Bi and In in Sn matrix are quite limited, where are Bi and In enrichment phases? The author need to provide more information about the influence of the Ag_3Sn and Cu_6Sn_5 IMCs and the Bi and In enrichment phases on the nucleation of β -Sn.
3. It is suggested to use [100] and [001] rather than $\langle 100 \rangle$ and $\langle 001 \rangle$ in Figs. 1, 2, 5 and 6.

Response to comments:

Reviewer text is in blue italics, our response is in black.

We are grateful to the Reviewers for raising valuable comments and suggestions. Based on the review, we have made revisions (highlighted in yellow in the revised manuscript) that we feel have improved the manuscript.

Reviewer #1

This paper presents a major step forward for improving the reliability and design of solder joints. It could well be the start of a revolutionary change in the industry. The authors have recognized the importance of oriented nucleation and found a practical means to get this under control, with impressive results. There are minor editorial suggestions in the attached document, but the paper is publishable in NCOMMS.

p. 5 line 21 electronic fluxes, not electronics fluxes

The 'electronics fluxes' has been changed into 'electronic fluxes' on page 5 line 21.

The reason for the positioning of the black square in Figure 2A is not apparent.

To clarify this, the following text has been added to the caption of Figure 2 on page 22:

The black square on each net of Sn atoms indicates the projection of the corresponding unit cell, and its position is determined by the origin for the crystallographic settings in Table 1.

p. 6: How were Sn droplets made and deposited on the substrate? This is not described. Were they deposited as balls, and then heated and cooled in a DSC?

Further details of the droplet nucleation experiments have been added to the Methods section on page 13:

'99.9% purity Sn particles (balls) of between 1 and 50 μ m were placed on the largest facets of the as-extracted (without further processing) PtSn₄, α CoSn₃, and β IrSn₄ single crystals with a NH₄Cl-ZnCl₂ based flux (Stay-Clean liquid flux, HARRIS). The Sn was melted on a hot plate with a peak temperature of 240 °C to ensure wetting and spreading on the facet, and then IMC crystals with Sn droplets spread on were ultrasonically bathed in ethanol to remove flux residues. Thermal cycles were then performed in a Mettler Toledo DSC 1 under a nitrogen atmosphere with a heating rate of 0.17 K/s, peak temperature 240 °C, time above the eutectic temperature of 180 s and a cooling rate 0.33 K/s.'

The following text is added on page 6 line 24 to alert the reader to the detailed methods at the end of the paper:

Experimental details are as given in the Methods section.

Figure 4 caption: Please indicate what the 'inert' substrate was in the text as well as the caption or top of part (B) of the figure (e.g. 'glass?').

The following text has been added to page 7 line 23: ...inert oxidised Al substrates.

Figure caption 4 on page 24 was also expanded with : All samples were measured on inert oxidised Al substrates.

Also, did you use oriented single crystals of all three of the Stannide IMC materials? This is not clear. There should be enough information so that others can reproduce your work.

All three stannide IMCs were used for droplet nucleation experiments and joints. More details have been added to clarify the whole experimental processes in the Methods section on page 13 line 8:

...the largest facets of the as-extracted (without further processing) PtSn₄, αCoSn₃, and βIrSn₄ single crystals with a NH₄Cl-ZnCl₂ based flux (Stay-Clean liquid flux, HARRIS). The Sn was melted on a hot plate with a peak temperature of 240 °C to ensure wetting and spreading on the facet, and then IMC crystals with Sn droplets spread on were ultrasonically bathed in ethanol to remove flux residues.

p. 8

Al on Al₃Ti₄₄). However, importantly, the ORs reported in Table 2 for all 182 of the droplets examined, irrespective of which OR formed, the [001] of tetragonal βSn was always in the plane of the largest facet of the IMC. Thus, these ORs are not only reproducible but also useful since, by controlling the

You describe facets of the IMC, but the reader has no way of know what the facet refers to. I thought you were using a polished single crystal, but reference to a facet makes this seem like there was a polycrystal. Please clarify what the condition of the substrate was?

Sorry for the lack of clarity here. To make this clear, we have added the following text to page 8 line 7:

Since the natural growth shape of primary αCoSn₃, PtSn₄, and βIrSn₄ crystals are thin plates (Figure 3 (A)-(C)) whose main facet is the desired nucleation plane, seed crystals were obtained by dissolving the matrix βSn and using the largest natural growth facet as a seed crystal without the need to take slices from a wafer of IMC.

βSn grain structure and orientation control in solder joints. To incorporate a PtSn₄, αCoSn₃, or βIrSn₄ seed crystal into ball-grid array (BGA) solder joints, they were first bonded to Cu pads using a form of transient liquid phase bonding (TLPB). An immersion tin coating was applied to Cu pads and the nucleant IMC wafer was laid flat with the main facet in the plane of the pad as shown in Figure 5 (A) using αCoSn₃ as an example. The αCoSn₃ was then TLPB to the pad by a reflow of 5 minutes at 240°C,

Please clarify: while you used TLPB, did the little wafer of the IMC stay solid during the bonding process? Was there any flux used?

To clarify this, text was added on page 8 line 12:

...where the IMC seed crystal remained solid during the TLPB.

Text was also added in the Methods section on page 14 line 7 to clarify what flux is used: PtSn₄, αCoSn₃ or βIrSn₄ crystals were placed on the tin layer with NH₄Cl-ZnCl₂ based flux (Stay-Clean liquid flux, HARRIS) coated on.....

Regarding the prior comment, it appears that you sliced wafers of a IMC single crystal and that you refer to the flat surface as the facet. Is this correct?

Sorry for the lack of clarity here. To make this clear, we have added the following text to page 8 line 7:

Since the natural growth shape of primary αCoSn₃, PtSn₄, and βIrSn₄ crystals are thin plates (Figure 3 (A)-(C)) whose main facet is the desired nucleation plane, seed crystals were

obtained by dissolving the matrix βSn and using the largest natural growth facet as a seed crystal without the need to take slices from a wafer of IMC.

p. 9 The nucleation undercooling of 550 μm Cu/SAC305+IMC/Cu joints for each type of IMC seed crystal were measured and are plotted versus the lattice registry in Figure 4 (D?). There is a similar trend

Practical question – did you put a board+chip with a single solder joint in the dsc? It is hard to imagine what you did to get the data in Figure 4D?

The methods section answers above questions adequately, except for the making of a BGA joint in a DSC.

To make this clear, additional text is added to the Methods section on page 14 line 13:

All solder joints were firstly made by reflowing in the Tornado LFR400 reflow oven. Some joints were reflowed subsequently in the DSC to measure the nucleation undercooling. Substrates were cut to $\sim 3 \times 3 \text{mm}$ for DSC measurements.

In the main body of the text, clarifying how you made and placed the nucleant wafers would help the reader not get distracted from your story.

The paragraph on page 8 is expanded with the new yellow text to clarify the joint-making process:

βSn grain structure and orientation control in solder joints. To incorporate a PtSn_4 , αCoSn_3 , or βIrSn_4 seed crystal into ball-grid array (BGA) solder joints, they were first bonded to Cu pads using a form of transient liquid phase bonding (TLPB), where the IMC seed crystal remained solid during the TLPB. An immersion tin coating was applied to Cu pads and the nucleant IMC was laid flat with the main facet in the plane of the pad as shown in Figure 5 (A) using αCoSn_3 as an example. The αCoSn_3 was then TLPB to the pad by a reflow of 5 minutes at 240°C , which resulted in the $\text{Cu}/\text{Cu}_3\text{Sn}/\text{Cu}_6\text{Sn}_5/\alpha\text{CoSn}_3$ layers shown in Figure 5 (B). Since PtSn_4 , αCoSn_3 and βIrSn_4 are all solderable surfaces using a standard ROL-1 flux, these nucleant-modified pads were then used in the same manner as a Cu-OSP substrate and solder joints were made following the procedure in the Methods section.

p. 12

The Sn was melted on a hot plate with a peak temperature of 240°C to ensure wetting and spreading on the facet, and then flux residues were removed. Thermal cycles were then performed 10

How were flux residues removed?

Sorry for the lack of this information here. Additional text has been added to the Methods section on page 13 line 10 to clarify this:

.....and then IMC crystals with Sn droplets spread on were ultrasonically bathed in ethanol to remove flux residues.

chemical analysis mould for X-Ray Fluorescence (XRF) spectroscopy analysis. To prepare solder balls with a diameter of $550 \pm 25 \mu\text{m}$, Sn-3.0Ag-0.5Cu and Sn-3.5Ag were rolled to 0.05 mm foils, punched to $\approx 1.6 \text{mm}$ discs, and reflowed on an inert hotplate with a ROL-1 tacky flux (Nihon Superior Co., 20 what was the inert surface?

Text has been added to page 13 line 21 and 22 to make this clear:

.....and reflowed on a non-wetting highly-oxidised Ni sheet with a ROL-1 tacky flux (Nihon Superior Co., Ltd.) and peak temperature of 280°C .

p. 13

crystals were placed on the tin layer with their main facet near-parallel with the substrate at a desired rotation angle, and the nucleants were transient liquid phase bonded to the substrate by holding at 240 °C for 5 min (for the αCoSn_3 case) or at 300 °C for 180 min (for the PtSn_4 and IrSn_4 cases). Since PtSn_4 , αCoSn_3 and βIrSn_4 are all solderable surfaces using a standard ROL-1 flux, these nucleant-modified pads were then used in the same manner as a Cu-OSP substrate.

What about environment – in air or nitrogen?

Text has been added to page 14 line 15 to make this clear:

...N₂ atmosphere was used in all reflowing cases.

Reviewer #3

Both academic and industrial people are eager to seek methods to effectively control the grain orientation of solder joints due to the anisotropy of β -Sn in mechanical and physical properties. The authors claim they have generated single crystal joints by using IMC nucleants as seed crystal. The authors are supposed to address the comments:

1. For Fig. 3, could the authors provide the EBSD map in X and Z directions? Similarly, the EBSD maps in X and Y directions for Figs. 5 and 6.

Maps in the other directions have been added to the Supplementary Information, and are also given below. The IPF X, Y and Z maps of the new Supplementary Figures 3, 4, 5, and 6 all show joints with βSn orientated with its c-axis near-parallel with the substrate plane. In Supplementary Figure 4, note that the mechanical twin across one sample and minor orientations at the sample edges were caused by deformation during sample preparation. We have also made all EBSD datasets available on the Zenodo repository for readers to explore if they wish. The following data availability statement has been added to the end of the Methods section:

Data availability. The EBSD and DSC datasets generated during the current study are available in the Zenodo repository (doi: 10.5281/zenodo.884113).

Supplementary Figure 1. A typical example of Sn droplets solidified on the (001) facet of βIrSn_4 and the corresponding EBSD maps. (A) The backscatter electron image. (B) IPF-X map. (C) IPF-Y map. (D) IPF-Z map. This is a supplementary figure for Figure 3.

Supplementary Figure 3. EBSD maps of the solder joint in Figure 5.

Supplementary Figure 4. EBSD maps of Cu/Sn-3Ag-0.5Cu or Sn-3.5Ag + αCoSn_3 /Cu solder joints in Figure 6.

Supplementary Figure 5. EBSD maps of Cu/Sn-3Ag-0.5Cu or Sn-3.5Ag + PtSn₄/Cu solder joints in Figure 6.

Supplementary Figure 6. EBSD maps of Cu/Sn-3Ag-0.5Cu or Sn-3.5Ag + βIrSn₄/Cu solder joints in Figure 6.

2. Several Sn-based solder alloys have been used in this study. Usually for Sn-Ag and Sn-Ag-Cu solders, Ag₃Sn and Cu₆Sn₅ IMCs are the primary phases rather than βSn dendrites during

a conventional reflow process. The precipitation of these IMCs also have a great effect on the nucleation of βSn . Did the author observe the formation of Ag_3Sn particles or plates in the Ag-bearing solder joints? If not, where is Ag? If yes, did the IMC phases still precipitate prior to βSn ? Also, since the solubilities of Bi and In in Sn matrix are quite limited, where are Bi and In enrichment phases? The author need to provide more information about the influence of the Ag_3Sn and Cu_6Sn_5 IMCs and the Bi and In enrichment phases on the nucleation of βSn .

We have added the following text to page 8 and 9 to provide more information on the influence of the Ag_3Sn and Cu_6Sn_5 IMCs:

The microstructure consists of primary Cu_6Sn_5 , βSn dendrites with $\langle 110 \rangle$ growth directions indicated by the blue arrows, $\beta\text{Sn} + \text{Ag}_3\text{Sn} + \text{Cu}_6\text{Sn}_5$ interdendritic eutectic, and a bonded αCoSn_3 particle on the bottom substrate.... Note that, even though primary and interfacial Cu_6Sn_5 are present prior to βSn nucleation, the nucleation of βSn always occurred on the seed crystal because they (αCoSn_3 , PtSn_4 , βIrSn_4) are more potent nucleants than Cu_6Sn_5 . Primary Ag_3Sn plates were not observed here because the seed crystals require only a relatively small undercooling for βSn nucleation. Further examples of the phases are shown in the Supplementary Information.

We have added a Supplementary Figure 2 to more clearly show the Ag_3Sn particles in the eutectic mixture and the primary Cu_6Sn_5 .

Supplementary Figure 2. Microstructure of a typical Cu/Sn-3Ag-0.5Cu or Sn-3.5Ag + IMC/Cu solder joint. (A) The macrostructure of a joint in which the seed crystal and numerous primary Cu_6Sn_5 phases can be seen. There are no primary Ag_3Sn phases. The microstructures shown **(B)** βSn dendrites and **(C)** $\beta\text{Sn} + \text{Cu}_6\text{Sn}_5 + \text{Ag}_3\text{Sn}$ ternary eutectic.

We have added the following text to page 12 to provide more information about the role of Bi and In enrichment phases on the nucleation of βSn .

The seed crystals remain effective with these additions because they do not react with the seed crystals or introduce a more potent nucleant phase. For example, the Ni addition mostly influences the Cu_6Sn_5 phase which is a less potent nucleant than the seed crystals;

and these Bi and In additions introduce extra (Bi) and $\zeta(\text{Ag,In})$ phases that form later during solidification and do not strongly affect the nucleation of βSn . The successful orientation control in these Ni, Bi and In-containing solders indicates that this orientation control method is likely to be applicable to solder compositions developed in the future.

We have also added further annotations to Supplementary Figure 7 to more clearly show the phases in each joint containing Ni, Bi and In.

Supplementary Figure 7. c-axis orientation control in joints of different alloys. The nucleant IMCs used are αCoSn_3 . From left to right, optical micrographs, EBSD IPF-Z maps, and micrographs shown phases in solder joints.

3. It is suggested to use [100] and [001] rather than <100> and <001> in Figs. 1, 2, 5 and 6.

Thanks for this suggestion. Direction indices in Figs. 1, 2, 5 and 6 are all changed to [100] and [001].

We hope that these changes are satisfactory and look forward to hearing from you.

Yours sincerely,

Zhaolong Ma and Chris Gourlay

REVIEWERS' COMMENTS:

Reviewer #3 (Remarks to the Author):

All the comments have been well responded. The revised version is pretty good and I will be happy to recommend its publication in Nature Communications.